# Approximate Birkhoff-von-Neumann decomposition: a differentiable approach

## Abstract

The Birkhoff-von-Neumann (BvN) decomposition is a standard tool used to draw permutation matrices from a doubly stochastic (DS) matrix. The BvN decomposition represents such a DS matrix as a convex combination of several permutation matrices. Currently, most algorithms to compute the BvN decomposition employ either greedy strategies or custom-made heuristics. In this paper, we present a novel differentiable cost function to approximate the BvN decomposition. Our algorithm builds upon recent advances in Riemannian optimization on Birkhoff polytopes. We offer an empirical evaluation of this approach in the fairness of exposure in rankings, where we show that the outcome of our method behaves similarly to greedy algorithms. Our approach is an excellent addition to existing methods for sampling from DS matrices, such as sampling from a Gumbel-Sinkhorn distribution. However, our approach is better suited for applications where the latency in prediction time is a constraint. Indeed, we can generally precompute an approximated BvN decomposition offline. Then, we select a permutation matrix at random with probability proportional to its coefficient. Finally, we provide an implementation of our method.

## 1 Introduction & Related work

Sampling from a doubly stochastic (DS) matrix is a significant problem that recently caught the attention of the machine learning community, with applications such as exposure fairness in ranking algorithms (Kahng et al., 2018; Singh & Joachims, 2018), strategies to reduce bribery (Keller et al., 2018; 2019), and learning latent representations (Mena et al., 2018; Grover et al., 2018; Linderman et al., 2018). We consider the Birkhoff-von-Neumann decomposition (BvND) (Birkhoff, 1946), which is deterministic and represents a DS matrix as the convex combination of permutations matrices (or permutation sub-matrices). In general, the BvND of a particular DS matrix is not unique. Sampling from a BvND boils down to selecting a sub-permutation matrix with a probability proportional to its coefficient. Current BvND algorithms rely on greedy heuristics (Dufossé & Uçar, 2016), mixed-integer linear programming (Dufossé et al., 2018), or quantization (Liu et al., 2018). Hence, these methods are not differentiable.

We rely on reparametrization techniques to use gradient-based algorithms (Grover et al., 2018; Linderman et al., 2018). Recently, Mena et al. (2018) introduced a reparametrization trick to draw samples from a Gumbel-Sinkhorn distribution. However, these methods can underperform in applications where there is a constraint in the prediction, as reparametrization methods require to solve a perturbed Sinkhorn matrix scaling problem.

In this work, we propose an alternative to Gumbel-matching-related approaches, which is well-suited for applications where we do not need to sample permutations online. Thus, our method is fast during prediction time by saving all components in memory. We call our algorithm: differentiable approximate Birkhoff-von-Neumann decomposition, and it is a continuous relaxation of the BvND. We rely on the recently proposed Riemannian gradient descent on Birkhoff polytopes. The main parameter is the number of components of the decomposition. We enforce an approximate orthogonality constraint on each component of the BvND. To our knowledge, this is the first gradient-based approximation of the BvND.

## 2  PRELIMINARIES

We first present some background on the BvND and comment on recent advances on Riemannian optimization in Birkhoff polytopes.

**Notations.**  We write column vectors using bold lower-case, e.g., $\mathbf{x}$. $\mathbf{x}_i$ denotes the $i$-th component of $\mathbf{x}$. We write matrices using bold capital letters, e.g., $\mathbf{X}$. $\mathbf{X}_{ij}$ denotes the element in the $i$-row and $j$-column of $\mathbf{X}$. $[n] = \{1, \ldots, n\}$. Letters in calligraphic, e.g. $\mathcal{P}$ denotes sets. $\| \cdot \|_F$ denotes the Frobenius norm of a matrix. $\mathbf{I}_p$ is the $p \times p$ identity matrix, and $\mathbf{1}_n$ is a 1 vector of size $n$. We use the superscript of a matrix $\mathbf{X}^l$ to indicate an element of a set. Thus, $\{\mathbf{X}^i\}_{i=1}^k$ be short for the set $\{\mathbf{X}^1, \ldots, \mathbf{X}^k\}$. However, we denote $\mathbf{X}^{(t)}$ a matrix $\mathbf{X}$ at iteration $t$. $\Delta_n$ denotes the $n-1$ probability simplex, and $\odot$ is the Hadamard product.

### 2.1  TECHNICAL BACKGROUND.

**Definition 1 (Doubly stochastic matrix (DS))** *A DS matrix is a non-negative, square matrix whose rows and columns sums to* 1*. The set of DS matrices is defined as:*

$$\mathcal{DP}_n := \left\{ \mathbf{X} \in \mathbb{R}_+^{n \times n} : \quad \mathbf{X}\mathbf{1}_n = \mathbf{1}_n, \, \mathbf{1}_n^\mathsf{T}\mathbf{X} = \mathbf{1}_n^\mathsf{T} \right\}. \tag{1}$$

**Definition 2 (Birkhoff polytope)** *The multinomial manifold of DS matrices is equivalent to the convex object called the Birkhoff Polytope (Birkhoff, 1946), an $(n-1)^2$ dimensional convex submanifold of the ambient $\mathbb{R}^{n \times n}$ with $n!$ vertices. We use $\mathcal{DP}_n$ to refer to the Birkhoff Polytope.*

**Theorem 1 (Birkhoff-von Neumann Theorem)** *The convex hull of the set of all permutation matrices is the set of doubly-stochastic matrices and there exists a potentially non-unique $\theta$ such that any DS matrix can be expressed as a linear combination of $k$ permutation matrices (Birkhoff, 1946; Hurlbert, 2008)*

$$\mathbf{X} = \theta_1 \mathbf{P}^1 + \ldots + \theta_k \mathbf{P}^k, \quad \theta_i > 0, \, \theta^\mathsf{T} \mathbf{1}_k = 1. \tag{2}$$

*While finding the minimum $k$ is shown to be NP-hard (Dufossé et al., 2018), by Marcus-Ree theorem, we know that there exists one constructible decomposition where $k < (n-1)^2 + 1$. Fig. 1 shows an illustration of the BvND.*

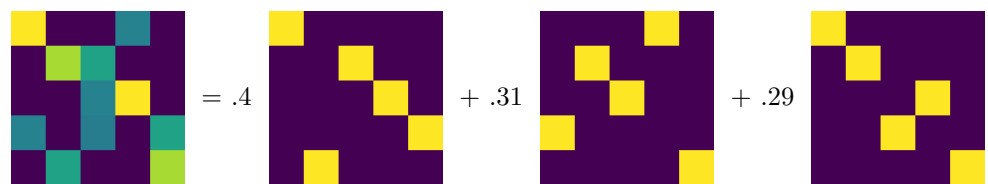

Figure 1: Illustration of the Birkhoff-von-Neumann decomposition (BvND): One can represent a doubly stochastic matrix as a convex combination of permutation matrices.

**Definition 3 (Permutation Matrix)** *A permutation matrix is defined as a sparse, square binary matrix, where each column and each row contains only a single true (1) value:*

$$\mathcal{P}_n := \left\{ \mathbf{P} \in \{0,1\}^{n \times n} : \quad \mathbf{P}\mathbf{1}_n = \mathbf{1}_n, \, \mathbf{1}_n^\mathsf{T}\mathbf{P} = \mathbf{1}_n^\mathsf{T} \right\}. \tag{3}$$

In particular, we have that the set of permutation matrices $\mathcal{P}_n$ is the intersection of the set of DS matrices and the orthogonal group $\mathcal{O}_n := \left\{ \mathbf{X}^\mathsf{T}\mathbf{X} = \mathbf{I} : \mathbf{X} \in \mathbb{R}^{n \times n} \right\}$, $\mathcal{P}_n = \mathcal{DP}_n \cap \mathcal{O}_n$ (Goemans, 2015).

**Riemannian gradient descent on $\mathcal{DP}_n$.**  The base Riemannian gradient methods use the following update rule $\mathbf{X}^{(t+1)} = R_{\mathbf{X}^{(t)}} \left( -\gamma H \left( \mathbf{X}^{(t)} \right) \right)$ (Absil et al., 2009), where $H \left( \mathbf{X}^{(t)} \right)$ is the Riemannian gradient of the loss $L : \mathcal{DP}_n \to \mathbb{R}$ at $\mathbf{X}^{(t)}$, $\gamma$ is the learning rate,

and $R_{\mathbf{X}^{(t)}} : T_{\mathbf{X}^{(t)}}\mathcal{DP}_n \to \mathcal{DP}_n$ is a retraction that maps the tangent space at $\mathbf{X}^{(t)}$ to the manifold. Douik & Hassibi (2019) defined $H\left(\mathbf{X}^{(t)}\right) = \Pi_{\mathbf{X}^{(t)}}\left[\nabla_{\mathbf{X}^{(t)}}L\left(\mathbf{X}^{(t)}\right) \odot \mathbf{X}^{(t)}\right]$, where $\Pi_{\mathbf{X}^{(t)}}$ is the projection onto the tangent space of DS matrices at $\mathbf{X}^{(t)}$. The total complexity of an iteration of the Riemannian gradient descent method on the DS manifold is $(16/3)n^3 + 7n^2 + \log(n)\sqrt{n}$ (Douik & Hassibi, 2019). See Appendix B for the closed-form computation of $\Pi_{\mathbf{X}^{(t)}}$ and $R_{\mathbf{X}^{(t)}}$.

# 3 Approximate Birkhoff-von-Neumann decomposition

We propose a differentiable loss function to approximate the BvND of a DS matrix. This approach is valuable when one can save the resulting decomposition in memory.

## 3.1 Formulation

Plis et al. (2011) showed that on $\mathcal{DP}_n$, all permutations are located on an hypersphere $\mathcal{S}^{(n-1)^2-1}$ of radius $\sqrt{n-1}$, $\mathcal{S}^{(n-1)^2-1} := \left\{\mathbf{x} \in \mathbb{R}^{(n-1)^2} : \|\mathbf{x}\| = \sqrt{n-1}\right\}$, centered at the center of mass $\mathbf{C}_n = \frac{1}{n}\mathbf{1}_n\mathbf{1}_n^\mathsf{T}$. Therefore, we can rely on the hypersphere-based relaxations (Plis et al., 2011; Zanfir & Sminchisescu, 2018) to learn the sub-permutation matrices. However, we have two main reasons to avoid the hypersphere relaxations in our setting: *i)* Birdal & Simsekli (2019) showed that the gap as a ratio between $\mathcal{DP}_n$ and both $\mathcal{S}^{(n-1)^2-1}$ and $\mathcal{O}_n$ grows to infinity as $n$ grows. *ii)* While there exist polynomial-time projections of the n!-element permutation space onto the continuous hypersphere representation and back, these algorithms are prohibitive in iterative algorithms, as each operation displays a time complexity of $O(n^4)$.

Another possible direction is to build a convex combination of sub-permutation matrices that is an $\epsilon$-close matrix of $\mathbf{M} \in \mathcal{DP}_n$. A $n \times n$ matrix $\mathbf{M}'$ is an $\epsilon$-close matrix of $\mathbf{M} \in \mathcal{DP}_n$ if $\left|\mathbf{M}_{ij} - \mathbf{M}'_{ij}\right| \leq \epsilon$, $\forall(i,j) \in [n]^2$. Barman (2015) showed that it is possible to build an $\epsilon$-close representation using at most $O(\log(n)/\epsilon^2)$ matrices. Kulkarni et al. (2017) further improve this result to using at most $1/\epsilon$ matrices. Therefore, we can build a matrix $\mathbf{M}'$ that satisfies $\|\mathbf{M} - \mathbf{M}'\|_\mathrm{F}^2 \leq \epsilon$, where $\mathbf{M}' = \sum_{l=1}^{k}\eta_l\mathbf{X}^l$ and $\mathbf{X}^l \in \mathcal{P}_n$ for $l \in [k]$. However, it is not currently practical to operate directly on $\mathcal{P}_n$ as we have to solve a combinatorial optimization problem. Additionally, $\mathcal{P}_n$ lacks a manifold structure. Thus, we relax the domain of the absolute permutations by assuming that each $\mathbf{X}^l \in \mathcal{DP}_n$ for $l \in [k]$ (Linderman et al., 2018; Birdal & Simsekli, 2019; Lyzinski et al., 2015; Yan et al., 2016). We add an orthogonal regularization to ensure that each matrix $\mathbf{X}^l$ is approximately orthogonal. We can use Riemannian gradient descent-based methods on the manifold of DS matrices to minimize the total loss, which has a time complexity of $O(n^3)$ per iteration.

## 3.2 Optimization problem

We want to solve the following optimization problem:

$$\min_{\eta \in \Delta^k} \frac{1}{2}\left\|\mathbf{M} - \sum_{l=1}^{k}\eta_l\mathbf{X}^l\right\|_F^2, \quad \text{s.t. } \mathbf{X}^l \in \mathcal{DP}_n \cap \mathcal{O}_n, \quad \text{for } l \in [k], \tag{4}$$

where $k$ is $O(1/\epsilon)$ (by Theorem 9 in Kulkarni et al. (2017)) for an $\epsilon$-close matrix approximation. Now, we recast Eq. 4 in its Lagrangian form and include additional penalties to leverage the Riemannian gradient descent on $\mathcal{DP}_n$.

**Reconstruction loss.** This loss measures the reconstruction error for a given set of candidate DS matrices, $\left\{\mathbf{X}^l\right\}_{l=1}^k$ and weight vector $\eta \in \Delta^k$, as follows:

$$L_{\mathrm{recons}}\left(\eta, \left\{\mathbf{X}^l\right\}_{l=1}^k\right) = \frac{1}{2}\left\|\mathbf{M} - \sum_{l=1}^{k}\eta_l\mathbf{X}^l\right\|_F^2. \tag{5}$$

**Orthogonal regularization.** This loss encourages a DS matrix $\mathbf{X}$ to be approximately orthogonal by pushing them towards the nearest orthogonal manifold (Brock et al., 2017). We compute it as follows:

$$L_{\text{ortho}}(\mathbf{X}) = \sum_{(i,j)\in[n]^2} \left| \left( \mathbf{X}^\mathsf{T}\mathbf{X} - \mathbf{I} \right)_{ij} \right|. \tag{6}$$

Eq. 6 corresponds to an entrywise matrix norm that promotes sparsity. Nevertheless, we can also use other orthogonality promoting losses like in Zavlanos & Pappas (2008) or Bansal et al. (2018), e.g., soft orthogonality regularization, $L_{\text{ortho}}(\mathbf{X}) = \|\mathbf{X}^\mathsf{T}\mathbf{X} - \mathbf{I}\|_F^2$. We note that $Ł_{\text{ortho}} : \mathcal{DP}_n \to \mathbb{R}^+$ and it is zero iff $\mathbf{X} \in \mathcal{O}_n$. However, Eq. 6 is not convex in $\mathbf{X}$ and we are guaranteed only to converge to saddle points.

**Optimization objective.** We compute the total loss by adding the reconstruction loss and an orthogonal regularization for each $\mathbf{X}^l$ for $l \in [k]$, as follows:

$$\min_{\left\{\mathbf{X}^l \in \mathcal{DP}_n\right\}_{l=1}^k, \eta \in \Delta^k} L\left(\eta, \left\{\mathbf{X}^l\right\}_{l=1}^k; \mu, \omega\right) = L_{\text{recons}}\left(\eta, \left\{\mathbf{X}^i\right\}_{l=1}^k\right) + \sum_{l=1}^k \mu_l L_{\text{ortho}}\left(\mathbf{X}^l\right) + \omega\,\Omega(\eta), \tag{7}$$

where $\Omega(\cdot)$ is a regularization function, $\omega > 0$ and $\mu_l > 0$ for $l \in [k]$ are regularization hyper-parameters that control the trade-off between reconstruction and orthogonality. For simplicity, we assume the same value of $\mu_l$ for all $l \in [k]$. Algorithm 1 displays a summary of our proposed solution. This algorithm has a complexity per iteration of $O(k\,n^3)$. Thus, we are interested in the $k \ll n$ regime.

**Diversity.** The regularization function $\Omega(\cdot)$ is necessary to avoid trivial solutions and force using various reconstruction components. For some $l \in [k]$, the model sets $\mathbf{X}^l$ to $\mathbf{M}$ if we do not regularize $\eta$. However, biasing the contribution of each component $\left\{\mathbf{X}^l\right\}_{l=1}^k$ is problem-dependent. We set $\Omega(\cdot) = \|\cdot\|_2^2$. We remark that this regularization does not impose fining different sub-permutation matrices. However, repeated sub-permutation matrices are not an issue for the BvND (see Fig. 3).

**Orthogonalization cost annealing (optional).** We can use a variable regularization hyper-parameter $\mu^{(t)}$ at training time to improve fining suitable solutions (Bowman et al., 2015). At the start of training, we set $\mu^{(0)} = 0$, so that the model learns to represent with various components. Then, as training progresses, we gradually increase this parameter, forcing the model to impose the orthogonality constraint. This process boils down to computing $\mu^{(t)} = \min\left(1, \max\left(0, \frac{t-t_i}{t_o-t_i}\right)\right)\mu$, were $t_i$ and $t_o$ denote the initial and final iteration of the annealing, respectively. However, we also need to tune $t_o$ and $t_i$.

**Refinement.** The primary use of the BvND is to sample (binary) permutations matrices from a DS matrix. However, the solution of Eq. 7 yields approximately orthogonal matrices. Thus, we still need to round/refine the solution to return a permutation matrix. We can use two different approaches: deterministic rounding or rounding by stochastic sampling. The deterministic rounding finds a feasible permutation of a given DS matrix via the Hungarian algorithm (optimal transport problem) (Peyré et al., 2019; Birdal & Simsekli, 2019), where we set the (transport) cost for each $l \in [k]$ to $\mathbf{K}^l = 1 - \mathbf{X}^l$. Thus, we solve $\bar{\mathbf{X}}^l \leftarrow \min_{\bar{\mathbf{X}}^l \in \mathcal{P}_n} \frac{1}{n}\sum_{i=1}^n \mathbf{K}^l_{i,\bar{\mathbf{X}}^l_i}$, where $\bar{\mathbf{X}}^l_i$ denotes the nonzero column of the $i$th row of the matrix $\bar{\mathbf{X}}^l$. However, the matching is not differentiable. In that case, we can also use the Sinkhorn algorithm (Cuturi, 2013), which solves an entropic-regularized optimal transport problem with cost $\mathbf{K}^l$. We set a small entropic regularization (temperature) parameter. Our experiment showed similar performance for the matching and Sinkhorn, which suggests that our approximation can result in practical end-to-end applications.

---

**Algorithm 1** Approximate BvND with a differentiable cost function

---

**Require:** DS matrix $\mathbf{M}$, number $k$ of sub-permutation matrices, learning rate $\gamma$, regularization parameters $\mu$ and $\omega$

**Ensure:** $\left(\eta^*, \left\{\mathbf{X}^{i*}\right\}_{i=1}^k\right)$ that minimizes Eq. 7

 1: **while** not converge **do**
 2:    **for** $i \in [k]$ **do**
 3:       Update using Riemannian gradient descent:

$$\mathbf{X}^{i(t+1)} \leftarrow R_{\mathbf{X}^{i(t)}}\left[-\gamma\,\Pi_{\mathbf{X}^{i(t)}}\left[\nabla_{\mathbf{X}^{i(t)}} L\left(\mathrm{softmax}(\eta^{(t)}), \left\{\mathbf{X}^{i(t)}\right\}_{i=1}^k ; \mu^{(t)}, \omega\right) \odot \mathbf{X}^{i(t)}\right]\right]$$

 4:    **end for**
 5:    Update using gradient descent:

$$\eta^{(t+1)} \leftarrow \eta^{(t)} - \gamma\nabla_{\eta^{(t)}} L\left(\mathrm{softmax}(\eta^{(t)}), \left\{\mathbf{X}^{i(t)}\right\}_{i=1}^k ; \mu^{(t)}, \omega\right)$$

 6: **end while**

---

## 4   APPLICATION: FAIRNESS OF EXPOSURE IN RANKINGS

One recent application of the BvND in machine learning is the reduction of presentation bias in ranking systems Singh & Joachims (2018). The aim is to find a DS matrix representing a probabilistic ranking system that satisfies a fairness constraint in expectation. Then, we sample a ranking from this fair DS matrix for each user. Thus, sampling for each user with Gumbel-Shinkhorn becomes prohibitive as it solves a Sinkhorn matrix scaling problem for each sample. Therefore, we precompute the BvND and propose rankings at random for each user. Here, $|\cdot|$ denotes the cardinality of a set.

Learning to rank algorithms tends to display top-ranked results more often given the user feedback (usually clicks), which leads to ignoring other potentially relevant results. In brief, the method finds a probabilistic re-ranking system which satisfies specific presentation bias constraints We encode this probabilistic re-ranking in a DS matrix. We decompose this DS matrix using the BvN algorithm. Then, one can select a sub-permutation matrix at random using a probability proportional to the decomposition coefficient. Thus, this re-ranking approach will satisfy group fairness constraints in expectation.

For simplicity, we assume a single query $\mathbf{q}$ and consider that we want to present a ranking of a set of $n$ documents/items $\mathcal{D} = \{\mathbf{d}_i\}_{i=1}^n$. We denote by $\mathcal{U}$ the set of all users $\mathbf{u}$ that lead to identical $\mathbf{q}$. We represent $\mathrm{rel}(\mathbf{d}, \mathbf{q})$ the measure of relevances for a given query[1]. We assume a full information setting. Thus relevances are known. We use the relevances to compute the utility, e.g., discounted cumulative gain (DCG), or the Normalized DCG (NDCG, the DCG normalized by the DCG of the optimal ranking) (Järvelin & Kekäläinen, 2002).

### 4.1   STATIC CASE

We can write a utility function $U$ in terms of a probabilistic ranking $\mathbf{M}$ for a query $\mathbf{q}$ as

$$U(\mathbf{M}|\,\mathbf{q}) = \sum_{\mathbf{d}_i \in \mathcal{D}} \sum_{j=1}^n \mathbf{M}_{ij}\, u(\mathbf{d}_i|\,\mathbf{q})\, v(j), \tag{8}$$

where $\mathbf{M} \in \mathcal{DP}_n$ is the probabilistic ranking. Thus $\mathbf{M}_{ij}$ represents the probability of replacing document $\mathbf{d}_i$ (indexed at position $i$) at rank $j$. $u(\mathbf{d}|\,\mathbf{q}) := \sum_{\mathbf{u}\in\mathcal{U}} \Pr[\mathbf{u}|\,\mathbf{q}]\, f(\mathrm{rel}(\mathbf{d}, \mathbf{q}))$, is the expected utility of document $\mathbf{d}$ for query $\mathbf{q}$, where $f(\cdot)$ maps the relevance of the document for a user to its utility. $v(j)$ is the examination propensity; it models how much attention an item $\mathbf{d}$ at position $j$. We use a logarithmic discount in the position $\mathbf{v}_j = v(j) = \frac{1}{\log_2(1+j)}$

---

[1]We can extend this definition to include user preferences, $\mathrm{rel}(\mathbf{d}, \mathbf{q}, \mathbf{u})$.

and $f(\text{rel}(\mathbf{d}, \mathbf{q})) = 2^{\text{rel}(\mathbf{d}, \mathbf{q})} - 1$. Thus, the utility function Eq. 8 corresponds to the expected DCG.

We include fairness constraint by solving: $\mathbf{M}^* = \arg\max_{\mathbf{M} \in \mathcal{DP}_n} U(\mathbf{M}|\mathbf{q})$ subject to $\mathbf{M}$ is fair. Singh & Joachims (2018) define several fairness constraints that are linear in $\mathbf{M}$, and this problem boils down to solving a linear program. However, in prediction time, one needs to sample from $\mathbf{M}$. Thus, we use the BvN algorithm to decompose $\mathbf{M}$ into the convex combination of $k$ permutation matrices $\mathbf{P}^k$. One chooses a permutation at random with a probability proportional to the coefficient. The resulting model satisfies the fairness constraint in expectation.

To set our fairness constraints, we need first to define the merit, impact, and exposure of an item $\mathbf{d}$. The merit of item is its expected average relevance. The exposure of item $\mathbf{d}$ is the probability $P(\mathbf{d})$ that the user will see $\mathbf{d}$ and thus can read that article, buy that product, etc. We note that estimating the position bias is not part of our study. We assume full knowledge of these position-based probabilities. We use the feedback $C$, e.g., clicks, as a measure of impact. We extend these definitions to group $\mathcal{G} \subseteq \mathcal{D}$ by aggregating over the group. Then, we have for a protected group $\mathcal{G}$

$$\text{Merit}(\mathcal{G}) = \frac{1}{|\mathcal{G}|} \sum_{\mathbf{d} \in \mathcal{G}} u(\mathbf{d}|\mathbf{q}), \quad \text{Imp}(\mathcal{G}) = \frac{1}{|\mathcal{G}|} \sum_{\mathbf{d} \in \mathcal{G}} C(\mathbf{d}), \text{ and } \text{Expo}(\mathcal{G}) = \frac{1}{|\mathcal{G}|} \sum_{\mathbf{d} \in \mathcal{G}} P(\mathbf{d}). \quad (9)$$

Setting the constraints as functions of the probabilistic ranking $\mathbf{M}$. $\hat{\text{Expo}}(\mathbf{d}_i|\mathbf{M}) = \sum_{j=1}^{n} \mathbf{M}_{ij} \mathbf{v}_j$. Assuming the Position-Based Model (PBM) click model, the estimated probability of a click is the exposure $\times$ conditional relevancy. Thus, the estimated probability of click on a document $\mathbf{d}_i$ is $C(\mathbf{d}) = \hat{\text{Expo}}(\mathbf{d}|\mathbf{M}) u(\mathbf{d}|\mathbf{q})$. Therefore, we can estimate the average impact and exposure on the items in group $G$ for the rankings defined by $\mathbf{M}$ as

$$\hat{\text{Imp}}(\mathcal{G}|\mathbf{M}) = \frac{1}{|\mathcal{G}|} \sum_{\mathbf{d}_i \in \mathcal{G}} u(\mathbf{d}|\mathbf{q}) \left( \sum_{j=1}^{n} \mathbf{M}_{ij} \mathbf{v}_j \right) \text{ and } \hat{\text{Expo}}(\mathcal{G}|\mathbf{M}) = \frac{1}{|\mathcal{G}|} \sum_{\mathbf{d}_i \in \mathcal{G}} \sum_{j=1}^{n} \mathbf{M}_{ij} \mathbf{v}_j. \quad (10)$$

Here, we only use disparate exposure and impact constraints as fairness constraints. The disparity constraints $D(\mathcal{G}_i, \mathcal{G}_j)$ is the difference of the fairness metric between protected groups $\mathcal{G}_i$ and $\mathcal{G}_j$, divided by their respective merit. We denote $D^E(\mathcal{G}_i, \mathcal{G}_j)$ and $D^I(\mathcal{G}_i, \mathcal{G}_j)$ to be the Disparity Exposure Constraint and Disparity Impact Constraint, respectively. Let $\hat{D}(\mathcal{G}_i, \mathcal{G}_j)$ be the difference in estimations.

## 4.2 Fairness in Dynamic Learning-to-Rank

Morik et al. (2020) present an extension of the fairness of exposure (see Section 4) to a dynamic setting. They propose a fairness controller algorithm that ensures notions group fairness amortized through time. This algorithm dynamically adapts both utility function and fairness as more data becomes available.

Here, we assume that both the exposure and the impact vary through time, $\text{Imp}_t(\mathcal{G})$, and $\text{Expo}_t(\mathcal{G})$. Then, we use the cumulative fairness constraint over $\tau$ time steps, e.g., $\frac{1}{\tau} \sum_{t=1}^{\tau} \text{Imp}_t(\mathcal{G})$. We extend it to their estimations too.

The optimization problem is

$$\mathbf{M}^* = \arg\max_{\mathbf{M} \in \mathcal{DP}_n, \zeta_{ij} \geq 0} U(\mathbf{M}|\mathbf{q}) - \lambda \sum_{ij} \zeta_{ij}$$

$$\text{s.t. } \forall \mathcal{G}_i, \mathcal{G}_j : \hat{D}_\tau(\mathcal{G}_i, \mathcal{G}_j) + D_{\tau-1}(\mathcal{G}_i, \mathcal{G}_j) \leq \zeta_{ij}, \quad (11)$$

where $\lambda \geq 0$ controls the trade-off between ranking score and fairness.

## 5 EXPERIMENTS

We explore our differentiable approximate BvND's behavior on synthetic data and validate its usefulness in the fairness of exposure in ranking problems. We aim to check its performance compared to the greedy combinatorial construction.

**Optimization.** We build DS Random matrices naively to initialize each component $\mathbf{X}^l$. For $l \in [k]$, we sample each element of matrix $\mathbf{M}_{ij}^l, \forall (i,j) \in [n]^2$ from a half-normal distribution, i.e., the absolute value of an i.i.d. sample drawn from a Gaussian. Then, we project onto DS using the Sinkhorn algorithm. We set the maximum number of iterations to $10\,000$, the learning rate $\gamma = 10^{-3}$, and relative tolerance of $10^4$. After a greedy parameter tunning, we use $\omega = 1$ and $\mu = 10^{-2}$.

**Technical aspects.** We give a Pytorch (Paszke et al., 2019) implementation in Appendix A. We use RiemannianADAM implemented in the geoopt library (Kochurov et al., 2020). We run all the experiments on a single desktop machine with a 4-core Intel Core i7 2.4 GHz.

### 5.1 SYNTHETIC DATA

**Setup.** We build a set of ten DS Random matrices $\{\mathcal{DP}_n\}_{n=1}^{10}$, where $n \in [4, 6, 10]$. To add sparsity, we mask each matrix by thresholding at $[0.1, 0.5, 0.9]$ divided $n$, respectively. Then, we project the masking result using the Sinkhorn algorithm. We explore the performance of the differentiable BvND as a function of the number of $k$ components. We measure the computation time, the reconstruction error, and the orthogonalization error.

| $n$ | $k$ | computation time (min) |
|---|---|---|
| 4 | 20 | $4.43 \pm 1.28$ |
| 6 | 30 | $8.68 \pm 2.18$ |
| 10 | 40 | $15.13 \pm 5.49$ |

Table 1: Computation time

**Results.** We observe in Fig. 2 the reconstruction error is monotonically decreasing. However, the approximate orthogonalization constraint becomes more challenging to satisfy when increasing the DS input matrix's dimensionality. Table 1 shows the the computation time for each parameter.

### 5.2 STATIC FAIRNESS OF EXPOSURE

**Setup.** We use the toy example presented in Singh & Joachims (2018). These data represent a web-service that connects employers (users) to potential employees (items). The set contains three males and three females. The male applicants have relevance for the

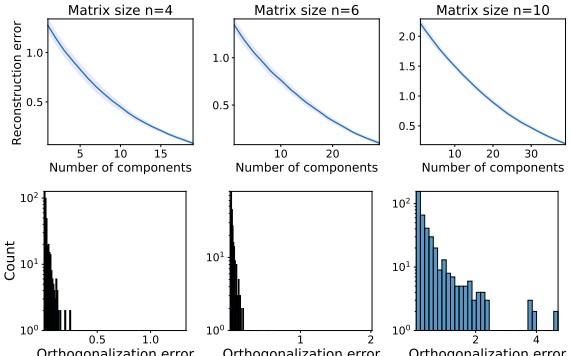

Figure 2: Performance of the approximate BvND on synthetic data: for different matrix sizes. *(top)* Reconstruction error as a function of the number of components. *(bottom)* Histogram of the orthogonalization error of each component.

employers of 0.80, 0.79, 0.78, respectively, while the female applicants have the relevance of 0.77, 0.76, 0.75, respectively. Here we follow the standard probabilistic definition of relevance, where 0.77 means that 77% of all employers issuing the query find that applicant relevant. The Probability Ranking Principle suggests ranking these applicants in the decreasing order of relevance, i.e., the three males at the top positions, followed by the females. The task is to re-rank them so that the system satisfies equal opportunity of exposure across groups. Thus, we solve a linear program to maximize Eq 8 such that $\hat{D}^E(\mathcal{G}_{\text{Male}}, \mathcal{G}_{\text{Female}}) \leq 10^{-6}$.

**Results.** Fig. 3 shows a toy example of the fairness of exposure. Fig. 3a presents the original biased ranking, whereas Fig. 3b shows a fair probabilistic ranking, which has a negligible loss in performance. Note that solving this problem does not imply that each

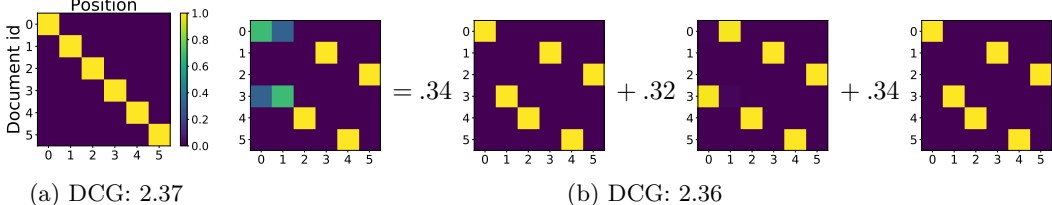

Figure 3: Static fairness exposure on toy data: *a)* Unfair ranking; *b)* Ranking satisfies the disparate exposure constraint. We decompose a fair probabilistic ranking using the approximate and differentiable BvND using three components.

sub-permutation matrix is unique. In practice, we often find a decomposition with repeated sub-permutations.

### 5.3 Dynamic Fairness of Exposure

**Setup.** We rely on Morik et al. (2020) to simulate an environment based on articles in the Ad Fontes Media Bias dataset, which generates a dynamic ranking on a set of news articles belonging to two groups left-leaning and right-leaning news articles, $\mathcal{G}_{\text{left}}$ and $\mathcal{G}_{\text{right}}$, respectively. See Appendix C for a full description of this simulation. We use dynamic learning to rank settings to minimize amortized fairness disparities. We solve Eq. 11 using linear programming. We evaluate the effectiveness of our differentiable approximate BvND (Diff-BvN) algorithm compared to the standard implementation of the BvND (BvN). We explore the difference between both models over various trade-offs between ranking score and fairness $\lambda$. Then, we measure their performance for a fixed $\lambda$.

**Baseline.** We use as a baseline the greedy heuristic of BvN (Dufossé & Uçar, 2016; Dufossé et al., 2018). For the Diff-BvN, we set the number of components to $k = 10$. We refine the matrix using the Hungarian algorithm (Kuhn, 1955) to ensure returning a permutation matrix. However, we also tried the stabilized Sinkhorn used in Cuturi et al. (2019) with the same performance.

**Results.** Fig. 4 presents the performance of BvN and Diff-BvN over various values of the fairness regularization parameter $\lambda$. Fig. 4a shows the NDCG of both methods, whereas Fig. 4c shows their unfairness of impact. Regarding the NDCG, BvN and Diff-BvN display the same performance across different values of $\lambda$. We observe the same pattern in the unfairness of exposure and impact, Fig.4b and Fig.4c, respectively.

We set the fairness regularization parameter to $\lambda = 10^{-2}$. We see in Fig. 5 that the performance of the re-ranked system is the same for BvN and Diff-BvN. However, the similar performance between these methods implies that only components 10 represent most of the information, which might not hold in other scenarios. Nevertheless, fewer components improve the performance in some applications (Porter et al., 2013; Liu et al., 2015).

## 6 Discussion and Conclusion

In this paper, we proposed a differentiable cost function to approximate the Birkhoff-von-Neumann decomposition (Diff-BvN). Our algorithm approximates a DS matrix by a convex combination of matrices on the Birkhoff polytope, where each matrix in the decomposition is approximately orthogonal. We can minimize the final loss function using Riemannian gradient descent. Our algorithm is easy to implement on standard auto-diff-based libraries. Experiments on the fairness of exposure in ranking problems show that our algorithm yields similar results to the Birkhoff-von-Neumann decomposition algorithm with a greedy heuristic. Our algorithm provides an alternative to existing approaches to sample permutation matrices from a DS matrix. In particular, it offers the option to balance the trade-off between memory and time in prediction settings. Fewer assignments lead to improved performance in some

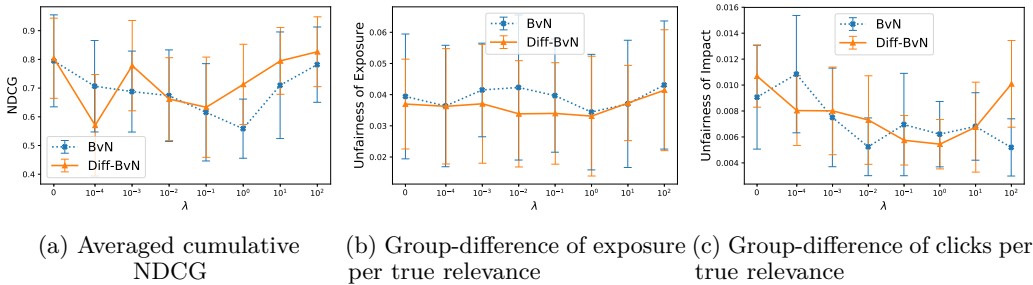

(a) Averaged cumulative NDCG

(b) Group-difference of exposure per true relevance

(c) Group-difference of clicks per true relevance

Figure 4: Performance of the fairness controller as a function of the parameter $\lambda$: Both methods, BvN and Diff-BvN display almost the same behavior across values of the regularization parameter. These values correspond to ten trials of the simulated news data of 3000 users.

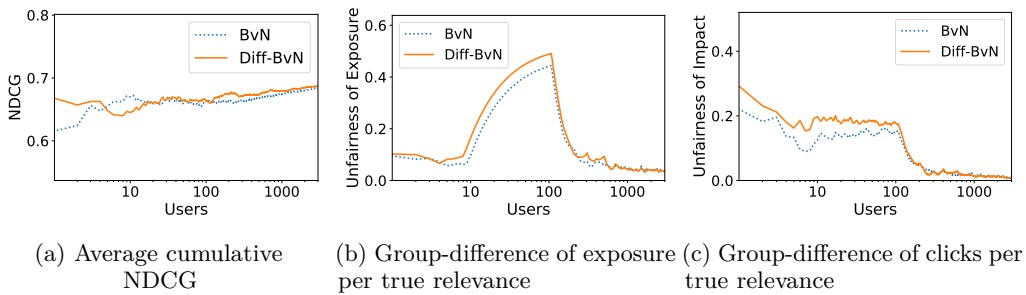

(a) Average cumulative NDCG

(b) Group-difference of exposure per true relevance

(c) Group-difference of clicks per true relevance

Figure 5: Performance of the fairness controller ($\lambda = 0.01$) as a function of the number of users: Both methods, BvN and Diff-BvN display the same convergence behavior on various metrics computed on ten trials of the simulated news data (3000 users).

applications (Porter et al., 2013; Liu et al., 2015). Thus, our algorithm can display better performance that greedy approaches as one sets the number of assignments a priory.

**Potential improvements.** In practice, our algorithm is sensitive to the orthogonal constraints. Therefore, we need to explore setting the parameters of the cost scheduler/annealing for the orthogonal regularization parameter, e.g., how fast do we have to increase the orthogonal regularization parameter?. Additionally, we note that the current implementation of our algorithm is still limited to small matrices. Thus, we need to explore different possible directions to make it scalable, e.g., randomization or quadrature methods (Altschuler et al., 2019).

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

## A    Pytorch implementation

```python
import torch
import geoopt
from torch import nn
from geoopt import ManifoldParameter
from geoopt.manifolds import BirkhoffPolytope

class BirkhoffVonNeumannDecomposition(nn.Module):
    """ Approximate differentiable Birkhoff-von-Neumann decompsition
    of a doubly-stochastic matrix.

    Parameters
    ____________
    n : int, size of the matrix.

    n_components : int (optional), the number of coefficients in the
    approximation.

    Attributes
    ____________
    manifold : BirkhoffPolytope
    Matrices: ManifoldParameter, Matrix on the Birkhoff Polytope.
            Used to compute compute RiemannianADAM.
    weight : nn.Parameter, Tensor of coefficients.

    """
    def __init__(self, n, n_components=2):
        super(BirkhoffVonNeumannDecomposition, self).__init__()
        self.n = n
        self.n_components = n_components
        self.manifold_ = BirkhoffPolytope()
        self.Matrices = nn.ParameterDict(
            {
                str(ind):
                ManifoldParameter(
                    data=self.manifold_.random((n, n)),
                    manifold=self.manifold_)
                for ind in range(n_components)
            } )
        self.weight = nn.Parameter(data=torch.rand(1, 1, n_components))

    def forward(self, input=None):
        w = torch.softmax(self.weight, dim=-1)
        Ms = torch.cat([
            self.Matrices[str(choice)].unsqueeze(2)
            for choice in range(self.n_components)
            ], dim=2)
        return (Ms * w).sum(-1)
```

Listing 1: Approximate Birkhoff-von-Neumann decomposition (ApproxBvND)

```python
import torch
import numpy as np

def ortho_error(X):
    n_samples = len(X)
    return (X.T @ X - torch.eye(n_samples)).abs().sum()

def reconstruction_error(A, A_approx):
    return torch.norm(A - A_approx, p='fro').pow(2)

def train(model,
        A: torch.Tensor,
        lr: float=1e-2,
        max_iter: int=5000,
        reg_weights: float=1.,
        reg_ortho: float=.01,
        stop_thr: float=1e-4):
    """Training function.

    A :   Tensor, is the doubly-stochastic matrix to approximate.

    lr : float (optional), learning rate of the RiemannianADAM.

    max_iter :   int (optional), maximum number of iterations.

    reg_weights : float (optional), regularization parameter for the
    weigths.

    reg_ortho : float (optional), regularization parameter for the
    orthogonal constraints.

    stop_thr : float (optional), stopping threshold.

    """
    optimizer = RiemannianAdam(list(model.parameters()), lr=lr)
    vloss = [stop_thr]
    loop = 1 if max_iter > 0 else 0
    it = 0
    while loop:
        it += 1
        optimizer.zero_grad()
        with torch.enable_grad():
            orth_loss = torch.zeros(1)
            for param in model.parameters():
                if isinstance(param,
                            geoopt.tensor.ManifoldParameter):
                    orth_loss += ortho_error(param)
                else:
                    weight_reg = model.weight.pow(2).sum()

        A_recons = model()
        reconstruction_loss = reconstruction_error(
            A, A_recons)

        loss = (
            reconstruction_loss
            + reg_ortho * orth_loss
            + reg_weights * weight_reg
        )
        vloss.append(loss.item())
```

```
relative_error = (
    abs(vloss[-1] - vloss[-2]) / abs(vloss[-2])
    if vloss[-2] != 0 else 0.)

if ((it >= max_iter) or
    (np.isnan(vloss[-1])) or
    (relative_error < stop_thr)):
    loop = 0

loss.backward()
optimizer.step()
return model
```

Listing 2: Training utils

## B  RIEMANNIAN OPTIMIZATION

A Riemannian manifold is a smooth manifold $\mathcal{M}$ of dimension $d$ that can be locally approximated by an Euclidean space $\mathbb{R}^d$. At each point $\mathbf{x} \in \mathcal{M}$ one can define a $d$-dimensional vector space, the tangent space $T_\mathbf{x}\mathcal{M}$. We characterize the structure of this manifold by a Riemannian metric, which is a collection of scalar products $\rho = \{\rho(\cdot, \cdot)_\mathbf{x}\}_{\mathbf{x}\in\mathcal{M}}$, where $\rho(\cdot, \cdot)_\mathbf{x} : T_\mathbf{x}\mathcal{M} \times T_\mathbf{x}\mathcal{M} \to \mathbb{R}$ on the tangent space $T_\mathbf{x}\mathcal{M}$ varying smoothly with $\mathbf{x}$. The Riemannian manifold is a pair $(\mathcal{M}, \rho)$ (Sommer et al., 2020).

The tangent space linearizes the manifold at a point $\mathbf{x} \in \mathcal{M}$, making it suitable for practical applications as it leverages the implementation of algorithms in the Euclidean space. We use the Riemannian exponential and logarithmic maps to project samples onto the manifold and back to tangent space, respectively. The Riemannian exponential map, when well-defined, $\mathrm{Exp}_\mathbf{x} : T_\mathbf{x}\mathcal{M} \to \mathcal{M}$ realizes a local diffeomorphism from a sufficiently small neighborhood $\mathbf{0}$ in $T_\mathbf{x}\mathcal{M}$ into a neighborhood of the point $\mathbf{x} \in \mathcal{M}$.

**Riemannian gradient descent.**  The base Riemannian gradient methods (Bonnabel, 2013; Smith, 1994) use the following update rule $\mathbf{w}_{t+1} = \mathrm{Exp}_{\mathbf{w}_t}(-\gamma_t H(\mathbf{w}_t))$, where $H(\mathbf{w}_t)$ is the Riemannian gradient of the loss $L : \mathcal{M} \to \mathbb{R}$ at $\mathbf{w}_t$, and $\gamma_t$ is the learning rate. However, the exponential map is not easy to compute in many cases, as one needs to solve a calculus of variations problem or know the Christoffel symbols (Bonnabel, 2013). Thus, it is much easier and faster to use a first-order approximation of the exponential map, called a retraction. A retraction $R_\mathbf{w}(\mathbf{v}) : T_\mathbf{w}\mathcal{M} \to \mathcal{M}$ maps the tangent space at $\mathbf{w}$ to the manifold such that $d(R_\mathbf{w}(t\mathbf{v}), \mathrm{Exp}_\mathbf{w}(t\mathbf{v})) = O(t^2)$, this imposes a local rigidity condition that preserves gradients. Therefore, one can rely on the retraction to compute the alternative update $\mathbf{w}_{t+1} = R_{\mathbf{w}_t}(-\gamma_t H(\mathbf{w}_t))$ (Absil et al., 2009).

Douik & Hassibi (2019) introduced the following computation of the retraction mapping and the projection onto the tangent space of DS matrices. The proofs can be found in (Douik & Hassibi, 2019).

**Theorem 2** *The projection operator* $\Pi_\mathbf{X}(\mathbf{Y})$ *maps* $\mathbf{Y} \in \mathcal{DP}_n$ *onto the tangent space at* $\mathbf{X} \in \mathcal{DP}_n$ *,* $T_\mathbf{X}\mathcal{DP}_n$ *is written as*

$$\Pi_\mathbf{X}(\mathbf{Y}) = \mathbf{Y} - \left(\alpha \mathbf{1}^\mathsf{T} + \mathbf{1}\beta^\mathsf{T}\right) \odot \mathbf{X}, \tag{12}$$

*where* $\odot$ *is the Hadamard product,* $\alpha = \left(\mathbf{I} - \mathbf{X}\mathbf{X}^\mathsf{T}\right)^+ \left(\mathbf{Y} - \mathbf{X}\mathbf{Y}^\mathsf{T}\right)\mathbf{1}$*,* $\beta = \mathbf{Y}^\mathsf{T}\mathbf{1} - \mathbf{X}^\mathsf{T}\alpha$*, and* $(\cdot)^+$ *denotes the pseudo-inverse*

**Theorem 3 (Retraction)** *For a vector* $\zeta_\mathbf{X} \in T_\mathbf{X}\mathcal{DP}_n$ *lying on the tangent space at* $\mathbf{X} \in \mathcal{DP}_n$*, the first order retraction map* $R_\mathbf{X}$ *is given by*

$$R_\mathbf{X}(\zeta_\mathbf{X}) = \Pi\left(\mathbf{X} \odot \exp\left(\zeta_\mathbf{X} \oslash \mathbf{X}\right)\right), \tag{13}$$

*where* $\oslash$ *is the Hadamard division, the operator* $\Pi$ *denotes the projection onto* $\mathcal{DP}_n$*, efficiently computed using the Sinkhorn algorithm (Sinkhorn & Knopp, 1967).*

## C  SIMULATIONS

**News data.**  Morik et al. (2020) contain the description of this dataset. Nevertheless, we added it for completeness. In each trial, we sample a set of 30 news articles $\mathcal{D}$. For each article $\mathbf{d}$, the dataset contains a polarity value $\rho^{\mathbf{d}}$ that we rescale to the interval between $-1$ and $1$. We simulate the user polarities. We draw the polarity for each user from a mixture of two normal distributions clipped to $[-1,1]$, $\rho^{\mathbf{u}_t} \sim \text{clip}_{[-1,1]} \left( p_{\text{neg}} \mathcal{N}(-.5, .2) + (1 - p_{\text{neg}}) \mathcal{N}(.5, .2) \right)$, where $p_{\text{neg}}$ is the probability of the user to be left-learning (mean $-0.5$). We use $p_{\text{neg}} = 0.5$. Besides, each user has an openness parameter $o^{\mathbf{u}_t} \sim \mathcal{U}(.05, .55)$, indicating the breadth of interest outside their polarity.

We draw the true relevance from the Bernoulli distribution $\mathbf{r}_t(\mathbf{d}) \sim$ Berrnoulli $\left[ p = \exp \left( \frac{-(\rho^{\mathbf{u}_t} - \rho^{\mathbf{d}})^2}{2(o^{\mathbf{u}_t})^2} \right) \right]$. We use the Position-based click model (PBM (Chuklin et al., 2015)) to model user behavior, where the marginal probability that a user $\mathbf{u}_t$ examines an article $\mathbf{d}$ depends only on its position. The remainder of the simulation follows the dynamic ranking setup. At each time step $t$ a user $\mathbf{u}_t$ arrives at the system, the algorithm presents an unpersonalized ranking and the user provides feedback $\mathbf{c}_t$ according to $\mathbf{p}_t$ and $\mathbf{r}_t$. The algorithm only observes $\mathbf{c}_t$ and not $\mathbf{r}_t$.

