# OpenReview forum: "Approximate Birkhoff-von-Neumann decomposition: a differentiable approach"
_ICLR.cc/2021/Conference — Reject_

### Official Review · AnonReviewer2 · 2020-10-21
**Review: Recommend rejection but with low confidence.**

**Rating:** 4
**Confidence:** 1

**Review:**

### Summary
**Objective:** decompose doubly stochastic matrix into convex combination of permutation matrices.

**Approach:** recast problem as an optimization problem by using a reconstruction loss and an orthogonality loss. Optimize "permutation matrices"-part by using Riemannian gradient descent and optimize coefficients using normal gradient descent. Refine final solution by either deterministic rounding or rounding by stochastic sampling.

### Strengths
**[+]** Idea is easy to understand, and the explanation is fairly straight-forward to read.

**[+]** The math seems correct.

### Weaknesses

**[-]** While the BvN decomposition might be of interest to the ICLR community, I am concerned the suggested differentiable approach to attaining the BvN decomposition is not.

**[-]** It seems fairly straight-forward to recast the matrix decomposition problem as an optimization problem and use RGD and GD. It is possible I am being to harsh on this account, and I'll be happy to revise my conviction if the other reviewers disagree.

**[-]** There is no time complexity analysis of Algorithm 1.

**[-]** The method is only tested for $n\le 30$, I am concerned the algorithm would not scale to even $n=100$. It is not clear to me whether this method will have any practical impact for $n=30$, however, it is definitely possible I am wrong on this account.

### Recommendation: Rejection 4, but with low confidence.

**[-]** While the BvN decomposition might be of interest to the ICLR community, I am concerned the suggested differentiable approach to attaining the BvN decomposition is not.

**[-]** It seems fairly straight-forward to recast the matrix decomposition problem as an optimization problem and use RGD and GD. It is possible I am being to harsh on this account, and I'll be happy to revise my conviction if the other reviewers disagree.

I am not very confident in this review and open to changing my recommendation, in particular, I'll re-evaluate my opinion based on the comments from the other reviews.

### Questions and Concerns
Please note the questions below were posted before this review and the authors already responded to these questions.

---

Thanks for submitting your work to ICLR. I hope you can help me clarify a few potential misunderstandings on my behalf.

Question 1. Is it correct that you named the approach 'differentiable' because it approximates the BvND by minimizing a differentiable loss?
I ask because I thought, after reading the title, that you compute BvND while supporting gradient computations, similar to how PyTorch supports gradient computations through eigendecomposition.

Answer: Yes.

Question 2. On the bottom of page 2 I was confused by  *"... the discrete set of permutation matrices $P_n$ is the intersection of the convex set of doubly stochastic matrices *and the* $DP_n\cap O_n$, ... "*.
Do you mean that $P_n = DP_n \cap O_n$?

Answer: Yes.

Question 3. Is it correct that experiment 5.1 and 5.2 use matrices of size at most $10$? You mention that finding the smallest $k$ is NP-Hard. But if $n=10$ I am not sure why it wouldn't just be possible to brute-force the solution. One can usually brute-force 3SAT or TSP problems of size $n=10$. Am I am misunderstanding something? Are the constants too big or something?

Answer: See the posted answer for details.

Question 4. It is not clear how big the matrices are in experiment 5.3. I get that you attempt to compute a decomposition with $k=10$ permutation matrices, but I am not sure how big the matrices are. In the appendix you state that you sample $30$ news articles and the article states there are $3000$ users. How big are the matrices?

Answer: The largest matrices are of size $(30, 30)$.

I apologize for any misunderstandings and look forward to reading your clarification.

---

### Additional Feedback

Comment 1. In the Preliminaries section, you introduce a lot of notation. The ICLR style template attempts to standardize notation, and I believe most of your notation is the same as the ICLR style template.

Comment 2. I was confused by the last sentence on page 2, I think you can revise/clarify this sentence.

Comment 3. In section 4 you write ".. bias constraints We encode this ...", you are missing a period before "We".

### Ethics Disclaimer
I have no training in ethics and thus have no basis to review the ethical implications of "{static, dynamic} fairness of exposure". I can read and understand the mathematical formulation of the problem, but for all I know, the problem formulation could be non-sensual from an ethical perspective.

---

### Official Review · AnonReviewer3 · 2020-10-25
**review: Approximate Birkhoff-von-Neumann decomposition: a differentiable approach**

**Rating:** 4
**Confidence:** 3

**Review:**

The paper deals with the problem of finding an (approximate) Birkhoff von Neumann decomposition of a doubly stochastic matrix DS, i.e., given a matrix DS find a convex combination of permutation matrices that sum to the given matrix DS.

The main innovation of the paper is a novel algorithm, based on Riemannian optimization, to find such a decomposition.
This algorithm leads to a set of approximately orthogonal matrices, based on which a set of permutation matrices can be constructed in a second step, e.g., via rounding or using a Sinkhorn algorithm.

While the paper contains a few interesting aspects, I think there are a number of problems with it.

## Presentation
The structure of the overall presentation is somewhat difficult to follow.
For instance, the paper lacks a clear mathematical problem statement that gives a precise definition of what the problem is that is to be solved here; and this only becomes partially clear as the paper unfolds.

A few questions I found myself having when reading the paper initially were:
Do the authors want to find 1 BvN decomposition? Sample from (create multiple feasible) BvN decompositions for a particular doubly stochastic matrix?
What is the relation to the fairness in ranking problems and how does their algorithm help in this context?
Why do other methods "underperform" when there is a "constraint in the prediction" and their approach does not?
When and why would we not need to "sample permutations online" -- which is presumably the scenario when their algorithm is advantageous?

In addition there are a number of typos and mistakes which impede a smooth reading of the paper. Some examples:
p.2 "DS matrices is incident" -> should be "equivalent"? if not what does incident mean in this context?
p.2. the last paragraph on the page is somewhat confusing the way it is written (cf. the clarification question by one reviewer). The orthogonal group should presumably have the same symbol twice (\mathcal{O}_m).
Why introduce the Stiefel manifold, if it is never used afterwards? It would be enough to simply introduce the orthogonal group as the set of square matrices with orthonormal columns (X^T X = I_n)??

Re: section 3 and 4. I found these sections difficult to follow.
For instance, how is a hypersphere relaxation defined? Why is it relevant here, if it is not used?
As another example, it seems the Barman (2018) comes after Kulkarni (2017), so how does the (chronologically) earlier paper improve upon a later paper??
How should we set the parameters in the optimization problem? How do we actually construct the permutation matrices from the outputs of the optimization.
The authors simply list a few high level alternatives but no precise description is given.

Where does it save computation/memory as advertised in the introduction?
Personally, I think the application is quite interesting, but based on the presentation I simply cannot see the connection based on what the authors wrote.
I don't think the connection of section 4 to the BvN decomposition problem is understandable from the presentations of the authors. Why is the result the authors present relevant?

## Theoretical contribution
I think the strongest potential contribution here is theoretical, in that a new algorithm is introduced for a relevant problem.
However, the paper does deliver a more detailed analysis of the algorithm but mainly states an optimization problem.
For instance, issues such as time / memory complexity are not discussed in more detail.
As mentioned above the procedure to actually construct permutation matrices is not described at all. In particular a rounding procedure would break the differentiability, which presumably is one of the advantages of the new method?
As the algorithm is approximate: what about the approximation error?
What is the sensitivity of the algorithm w.r.t. to the hyperparameters and how do we set them?
How does it effect the diversity of the permutation matrices one obtains (as the authors hint, there might be an issue here -- at least some more detailed discussion could be given).

## Application
I found the descriptions of the application to be not detailed enough to understand how the results of the authors come into play here.
Neglecting this aspect, it also remains unclear to me why the algorithm of the authors provides an advantage here as it does not seem to perform better in the applications shown compared to the baseline?
I think this would have potentially ok, if the paper would have provided a stronger theoretical analysis. However, as it stands I think both the "theory" as well as the "application" part of the paper fall below acceptance standards.

---

### Official Review · AnonReviewer4 · 2020-10-28
**In this paper the authors propose an optimization formulation for the decomposition of doubly stochastic matrices in terms of permutation matrices, where the cost function is differentiable. The optimization is carried out in the optimization over manifolds setting, using a recent result that gives a Riemannian structure to the set of doubly stochastic matrices.**

**Rating:** 5
**Confidence:** 3

**Review:**

In this paper the authors propose an optimization formulation for the decomposition of doubly stochastic matrices in terms of permutation matrices, where the cost function is differentiable. The optimization is carried out in the optimization over manifolds setting, using a recent result that gives a Riemannian structure to the set of doubly stochastic matrices.

The writing of the paper is correct in general, although there are some points commented below.

The term "differentiable" in this setting is somehow confusing. I would add "differentiable cost function" or something that leaves no room for misunderstandings. I was reading the paper thinking that I would find something different regarding the differentiable part.

The proposed formulation and optimization method seem correct, and to me it is a promising starting point. From here, I would expect to see some theoretical results regarding convergence (at least partially, in the spirit of [Z]), or applications where its full potential can be seen.

Section 4, which is an application of the proposed method, constitutes a great part of the paper (maybe more than half of it). Of course, the application of a given method is extremely important, but in this case it shifts the focus of the paper. Besides that, the writing of this large section seems to be quite different from the previous parts. While the first part was relatively well written and easy to follow, this half is rough.

To me, the interesting part of the paper is the method, and it falls short, given the comments above.

I would usually write this under "minor comments", but in this case it shows a lack of care and attention.
I appreciate the notation paragraph, but it seems to correspond to another paper. The notation for cardinality is never used as such, but the same notation is used for the absolute value several times. The upper index in $X^i$ is never defined (it could index a set of matrices, or indicate rows/columns). The Hadamard operations and pseudo-inverse are never used. On the other hand, in Algorithm 1 there are undefined operators, such as $\Pi$ and $R_X$ (which I assume are projection and retraction).

Minor comments:
 - Please add punctuation to the equations, since they are part of the text (in Definition 1 for instance)
 - The last sentence in page 2 is weird.
 - In the last sentence of the first paragraph, Section 3.1, there is an extra "f" (of f O(n^4) )
 - typo: geootp -> geoopt
 - The bibliography is not consistent. Some authors with full name, some with initials, some in all caps


[Z] Zavlanos, M. M., & Pappas, G. J. (2008). A dynamical systems approach to weighted graph matching. Automatica, 44(11), 2817-2824.

---

### Author Response · Authors · 2020-11-16
**Comments and changes**

We thank the reviewers for their honest comments, which help us improve this work. We are glad they recognize the interest of the problem and our algorithm.
We corrected issues related to our algorithm's presentation and added various discussions in the manuscript's current version. In particular, we introduced a discussion on the orthogonalization cost annealing and the diversity of sub-permutation matrices. We explicitly comment on the complexity of the algorithm.
We included a brief description of Riemannian gradient descent to clarify our main algorithm's notation. We also extended the "Refinement" part to improve clarity.

We acknowledge that we do not leverage our approach's end-to-end capability in our experimental setting. We impose the fairness constraint on the DS matrix and not on the drawn samples. However, we aim to show that our approximation leads to the same performance as greedy-BvND methods, which we confirm by the experimental results. Thus, one can "plugin" it into other problems.
This method is simple; nevertheless, it is novel and of interest to a community working on sampling permutations. We recognize the scalability issue of the current implementation. However, we believe this work is a step towards a scalable algorithm.

---

### Decision · Program_Chairs · 2021-01-07
**Final Decision**

**Decision:**

Reject

**Comment:**

The paper explores the Birkhoff-von-Neumann decomposition in order to propagate gradients through a bi-partite matching. The task is very relevant to the community but the reviewers raised concerns both about the theory and the practice of the work. Unfortunately the work is not ready for publication at ICLR.